# Therapeutic Potentials of Reducing Liver Fat in Non-Alcoholic Fatty Liver Disease: Close Association with Type 2 Diabetes

**DOI:** 10.3390/metabo13040517

**Published:** 2023-04-04

**Authors:** Georgios Tsamos, Dimitra Vasdeki, Theocharis Koufakis, Vassiliki Michou, Kali Makedou, Georgios Tzimagiorgis

**Affiliations:** 1Division of Gastroenterology, Norfolk and Norwich University Hospital, Norwich NR4 7UY, UK; 2Division of Endocrinology and Metabolism and Diabetes Center, First Department of Internal Medicine, Medical School, Aristotle University of Thessaloniki, AHEPA University Hospital, 54636 Thessaloniki, Greece; 3Sports Medicine Laboratory, School of Physical Education & Sport Science, Aristotle University of Thessaloniki, 57001 Thessaloniki, Greece; 4Laboratory of Biological Chemistry, Medical School, Aristotle University of Thessaloniki, AHEPA University Hospital, 54636 Thessaloniki, Greece

**Keywords:** fatty liver disease, fibrosis, resolution of NAFLD, diabetes mellitus, lifestyle approaches, glucose-lowering drugs

## Abstract

Nonalcoholic fatty liver disease (NAFLD), the most widespread chronic liver disease worldwide, confers a significant burden on health systems and leads to increased mortality and morbidity through several extrahepatic complications. NAFLD comprises a broad spectrum of liver-related disorders, including steatosis, cirrhosis, and hepatocellular carcinoma. It affects almost 30% of adults in the general population and up to 70% of people with type 2 diabetes (T2DM), sharing common pathogenetic pathways with the latter. In addition, NAFLD is closely related to obesity, which acts in synergy with other predisposing conditions, including alcohol consumption, provoking progressive and insidious liver damage. Among the most potent risk factors for accelerating the progression of NAFLD to fibrosis or cirrhosis, diabetes stands out. Despite the rapid rise in NAFLD rates, identifying the optimal treatment remains a challenge. Interestingly, NAFLD amelioration or remission appears to be associated with a lower risk of T2DM, indicating that liver-centric therapies could reduce the risk of developing T2DM and vice versa. Consequently, assessing NAFLD requires a multidisciplinary approach to identify and manage this multisystemic clinical entity early. With the continuously emerging new evidence, innovative therapeutic strategies are being developed for the treatment of NAFLD, prioritizing a combination of lifestyle changes and glucose-lowering medications. Based on recent evidence, this review scrutinizes all practical and sustainable interventions to achieve a resolution of NAFLD through a multimodal approach.

## 1. Putting into Context the Silent Epidemic: The Rise of NAFLD

Before the middle of the last decade, the worldwide prevalence of non-alcohol fatty liver disease (NAFLD) was approximately 25% in adult individuals. NAFLD is currently defined as an ectopic accumulation of lipids in the liver in the absence of secondary causes or other etiologies of liver disease [1,2]. It is histologically classified into two types: nonalcoholic fatty liver, defined as 5% liver steatosis with no evidence of injury to hepatocytes and no evidence of fibrosis; and nonalcoholic steatohepatitis (NASH), characterized as 5% liver steatosis, including inflammation and injury to hepatocytes with or without fibrosis [3]. NASH is estimated to be present in up to 60% of patients with NAFLD confirmed by biopsy [1]. Interestingly, while NAFLD was formally described more than 40 years ago, it has only been recognized in recent years as an important risk factor for metabolic disorders, related to the increasing burden of non-communicable diseases (NCDs), such as type 2 diabetes mellitus (T2DM), obesity, cardiovascular disease, and malignancy [4,5]. All of the above independent risk factors for mortality and morbidity are associated with a high economic cost to health care systems.

In the 21st century, among other metabolic diseases, diabetes mellitus has become the main concern for societies and health care systems. In particular, T2DM is mentioned to affect 1 in 11 adults and up to 463 million people worldwide [6]. According to the most recent data from the International Diabetes Federation (IDF), 700 million people between the ages of 20 and 79 will live with T2DM in 2045, around the world [7]. This type of diabetes, previously known as adult-onset diabetes, is defined by the following criteria: random blood sugar test ≥ 200 mg/dL (or ≥11.1 mmol/L) in symptomatic individuals, fasting blood sugar test ≥ 126 mg/dL (or ≥7 mmol/L) in two separate tests, oral glucose tolerance test (OGTT) ≥ 200 mg/dL (or ≥11.1 mmol/L) after two hours, and glycated hemoglobin (HbA1c) ≥ 6.5% in two separate assessments [8]. Growing evidence suggests that there is a strong positive association between NAFLD and diabetes mellitus, and, more specifically, individuals with diagnosed NAFLD have a twofold increased risk of T2DM [9]. This can be explained by the fact that the risk factors for diabetes overlap with the factors that affect the accumulation of liver fat. Thus, robust evidence from several epidemiological studies has reported that the detection of NAFLD in its early stages could predict the subsequent development of incident diabetes mellitus [10,11].

### 1.1. Pathophysiology of NAFLD/NASH

The development of NAFLD is an intricate process and is not completely understood. The liver, as is widely known, promotes many functions through the mobilization, regulation, and storage of nutrients. Hepatocytes play an essential role as regulators of amino acid, triglyceride and lipoprotein metabolism, gluconeogenesis, and ketogenesis [12]. Interestingly, although the liver is not the primary organ for lipid storage, there can be a few conditions that can cause ectopic lipid accumulation in it. The main causes include excessive intake of dietary fat, de novo lipogenesis, and hepatic uptake of non-esterified fatty acids from serum [12,13]. Insulin resistance causes lipolysis of adipose tissue, leading to a higher concentration of non-esterified fatty acids in the circulation, which are taken up by the liver in a concentration-dependent manner [14]. In addition, other metabolic disturbances, such as hyperinsulinemia and hyperglycemia, promote increased liver conversion of carbohydrates into fatty acids through de novo lipogenesis [15]. It is worth noting that the expression and action of various enzymes involved in lipogenesis and fatty acid storage in triglycerides are increased in individuals with NAFLD [16]. These enzymes, including diacylglycerol acyltransferase, stearoyl-CoA desaturase, fatty acid synthase, ketohexokinase, and acetyl-CoA carboxylase, have been explored as potential therapeutic weapons against NAFLD.

In a small percentage of patients, NAFLD is associated with infectious pathologies that can lead to the appearance of liver steatosis, such as hepatitis C and the human immunodeficiency virus (HIV). In some cases, it is related to medications (e.g., glucocorticoids, tamoxifen, tetracycline, total parenteral nutrition, amiodarone, methotrexate, valproic acid, and vinyl chloride) and specific toxins or acquired/inherited metabolic diseases such as cachexia, lipodystrophy, or even gastrointestinal surgery [17,18].

The development of NASH has been reported to be divided into two phases. The first is fatty deposition in the liver combined with an increase in insulin resistance. The second phase is related to cellular and molecular changes, including primarily oxidative stress and oxidation of fatty acids through lipid peroxidation, hyperinsulinemia, energy homeostasis, variation in the extracellular matrix, and changes in immune system function [19,20]. Consequently, the development of insulin resistance is even more complex than it appears. Although increased fat mass and adipocyte differentiation play a key role in the development of insulin resistance, the relationship between glycemia and the resolution of existing NAFLD/NASH (defined by the absence of ultrasound criteria for NAFLD/NASH on repeated imaging) is still a knowledge gap.

### 1.2. Unravelling the Connection: Pathophysiological Links between NAFLD and Diabetes Mellitus

Robust evidence suggests that there is a strong and bidirectional relationship between the progression of NAFLD and T2DM. High levels of diacylglycerol or ceramides can affect liver–insulin signaling, leading to an abnormal increase in liver insulin resistance [21]. Furthermore, elevated concentrations of circulating transaminases are strongly associated with a future higher risk of T2DM [22]. Except for the above, patients with NAFLD and T2DM are at an increased risk of developing macrovascular and microvascular diseases such as cardiovascular and chronic kidney disease, which are the main causes of mortality in these individuals [23]. Due to the intricate inter-relationship and the high global prevalence of NAFLD and T2DM, targeting insulin sensitivity and hyperglycemia combined with weight loss and adopting a holistic approach to the treatment of metabolic disease in patients with NAFLD could prove advantageous. Therefore, it has been suggested that alleviating NAFLD and NASH should be considered as part of the therapeutic strategy in patients with T2DM [24].

It should be noted that several studies have shown that improvements in insulin sensitivity have been positively associated with histological improvements in NAFLD/NASH and regression of fibrosis [25]. In 2013, Sung et al. reported that the risk of T2DM decreased in patients with resolved NAFLD status, as NAFLD is a reversible condition in its early stages [26]. A meta-analysis, published in 2018, found that the severity of NAFLD is directly related to dysglycemia, having a significant impact on the future risk of developing T2DM [23]. However, more studies are needed to prove the causal relationship between the two conditions and to reveal the extent of the risk of T2DM caused by the variable stages of NAFLD.

The purpose of this narrative review is to summarize the current literature on modern therapeutic approaches to NAFLD in patients with and without diabetes, focusing on the targeting of metabolic disturbances. Another purpose is to discuss future potential treatments and knowledge gaps in the therapy of NAFLD.

## 2. Pathways to Healing: Navigating the Therapeutic Approach

The therapeutic quiver of NAFLD consists of several levels, of which lifestyle, pharmaceutical, and surgical approaches are the main treatments. A multimodal intervention with multiple aspects, such as lifestyle modification, weight loss, specific diets, and medication, is the most appropriate and holistic approach for most people with NAFLD.

### 2.1. Less Is More: Unlocking the Power of Conservative Treatment

#### 2.1.1. The Power of Lifestyle Modification in NAFLD

Robust evidence supports the crucial role of lifestyle changes as primary options for the treatment of NAFLD. These approaches, which include diet, exercise, or physical activity, mainly aim to control metabolic status [27]. Looking back in 2004, the World Health Organization (WHO) established that moderate intensity exercise improves not only physical and mental health, but also metabolic syndrome, T2DM, and cardiovascular disease, conditions that are inseparably related to NAFLD [28]. In the 21st century, physical activity is considered a pillar determinant of metabolic control and is recommended for NAFLD. Different types of exercise, such as high-intensity intermittent exercise, aerobic exercise, or resistance exercise, seem to have beneficial effects on fatty liver disease [29,30]. In 2017, Oh et al. found that high-intensity interval aerobic exercise, moderate-intensity continuous aerobic exercise, and resistance exercise were equally effective in decreasing liver fat content. However, only high-intensity interval aerobic exercise had a beneficial effect on restoring Kupffer cell function [31]. A randomized control trial by Zhang et al. indicated that after a 12-month active intervention, the two exercise groups (strong and moderate training) showed a significant reduction in intrahepatic triglyceride content (measured by proton magnetic resonance spectroscopy) compared to the control group [32]. Furthermore, several studies reported that resistance training leads to a reduction in liver fat of 4–47% independently of weight loss [33]. The mechanisms underpinning the reduction of hepatic fat deposition due to exercise reflect changes in insulin sensitivity and circulatory lipids. Exercise not only improves glycemic control but suppresses de novo lipogenesis and improves blood pressure levels in people with NAFLD [34,35,36]. Table 1 summarizes key studies on the effect of different exercises in patients with NAFLD.

#### 2.1.2. Shedding Pounds: The Science of Weight Loss and Calorie Restriction

Weight loss is a gold standard therapy for most patients with NAFLD and can regress liver disease, along with the reduction of cardiovascular disease and the risk of T2DM [46]. Some researchers support that a weight reduction of 10% is capable of inducing the resolution of NASH and improving fibrosis by at least one stage [47]. According to the guidelines of the American Association for the Study of Liver Diseases (AASLD), a weight loss of 5–10% in overweight or obese individuals and 3–10% in non-obese individuals with NAFLD is the primary objective of lifestyle interventions. In accordance with the above, there are also the National Institute of Health and Care Excellence (NICE) guidelines [48,49]. In addition, obesity, as a result of excess caloric consumption, is one of the leading factors for NAFLD. Caloric restriction acts on metabolic reprogramming and on the utilization of body energy, reducing oxidative damage to cells [50]. Because carbohydrates (which are the main energy source of the human body) are linked to NAFLD, their restriction in the diet can lead to lower glycemic load, increased insulin sensitivity, and pancreatic β-cell insulin secretion of pancreatic cells [51].

A clinical trial by Holmer et al., which recruited 74 patients with NAFLD, indicated that intermittent calorie restriction and a low carbohydrate high fat diet (LCHF) are more effective in reducing liver steatosis and body weight compared to general lifestyle modification. Participants were randomized into 3 groups: intermittent calorie restriction, including 500 kcal/day for women and 600 kcal/day for men; LCHF, with an average daily calorie intake of 1600 kcal/day for women and 1900 kcal/day for men; and general lifestyle advice [52]. Furthermore, in a prospective study by Vilar et al., a combination of exercise and a hypocaloric diet revealed a dose–response relationship between weight reduction and general histological parameters, with the greatest improvement detected in those with the greatest weight loss [53]. However, the beneficial effects of a low-carbohydrate diet are only in the short term. In the long term, a reduced carbohydrate and a reduced fat diet has results similar to those who achieved a 7% weight loss [54]. Table 2 presents the effect of dietary intervention on NAFLD.

In addition to the above, it should be noted that some studies reported the beneficial effect of diabetes remission on NAFLD and pancreatic morphology. In 2020, a post hoc analysis of the DiRECT trial showed changes in the gross morphology of the pancreas 2 years post T2DM remission. The size of the pancreas had increased in patients who achieved remission and weight loss, compared to those who did not respond to the weight loss intervention. Intrahepatic fat and levels of FGF-21 and FGF-19 also decreased. However, it is notable that there is no significant increase in pancreas volume after 6 months of reversal of type 2 diabetes [55]. Additional trials might be of interest from a scientific point of view to investigate further data on the progression of NAFLD and changes in pancreatic tissue after remission of diabetes.

#### 2.1.3. Breaking the Link between Fatty Liver and Type 2 Diabetes: The Power of Nutritional Interventions

Numerous studies have corroborated the pivotal role of certain macronutrients in the initiation and progression of NAFLD, regardless of caloric intake. In particular, macronutrients such as saturated fatty acids (SFA), trans fats, simple sugars such as sucrose and fructose, and animal proteins are known to inflict damage on the liver through the accumulation of triglycerides and impaired antioxidant activity, compromising insulin sensitivity and postprandial triglyceride metabolism [56]. In contrast, the consumption of monounsaturated fatty acids (MUFA), ω3 polyunsaturated fatty acids (PUFA), plant-based proteins, and dietary fibers such as whole grain cereals, fruits and vegetables, fatty fish (which are primarily rich in ω3), and extra virgin olive oil have been found to confer beneficial effects [57,58]. Gupta et al. suggest that oily fish (2–4 g/d), coffee (≥3 cups/day), and nuts (100 g/d) are recommended as suitable additions to physical activity and caloric restriction for patients with fatty liver disease, based on strong evidence from human trials. Although tea, red wine, avocado and olive oil can be consumed moderately without harm, more research is needed to investigate their therapeutic benefits for patients with NAFLD/NASH [59].

Furthermore, Halima et al. conducted a study investigating the impact of apple cider vinegar on rats with diabetes and demonstrated that in addition to its potent antihyperglycemic properties, it also exhibited a crucial hepatoprotective effect. In particular, indicators of liver toxicity, namely ALT, AST, total and direct bilirubin, as well as levels of TC, TG, and LDL-c, demonstrated a significant reduction, which was particularly prominent after four weeks of treatment, together with an elevation in HDL-c [60]. These findings are consistent with several other studies [61,62,63]. The above elucidated results unequivocally demonstrate that daily ingestion of vinegar can mitigate the increase in blood glucose levels and lipid profile, which is typically induced by a hypercaloric diet in rats, as posited by Ousaaid et al. [64]. Therefore, the use of apple cider vinegar could confer considerable advantages in avoiding metabolic irregularities commonly associated with a high-calorie diet. 

**Table 2 metabolites-13-00517-t002:** Dietary interventions and outcomes in NAFLD patients.

Study	Type of Study	Number of Patients	Type of Diet	Patients with NAFLD	Duration	Insulin Resistance	Outcomes
Haufe et al. (2011) [65]	RCT	52 vs. 50	Hypocaloric LCD vs. LFD	Overweight/obese	6 months	↓ to a similar extent	42% vs. 47% ↓ in IHLC
Browning et al. (2011) [66]	Non-RCT	9 vs. 9	VLCD vs. Hypocaloric diet	Without cirrhosis	2 weeks	Not evaluated	55% vs. 28% ↓ in IHLC
Ryan et al. (2013) [67]	RCT	6 vs. 6	Med. vs. LFD or HCD	Biopsy-proven	6 weeks	↓ with the Med. diet	39% vs. 7% ↓ of IHLC
Vilar-Gomez et al. (2015) [53]	Single-arm	261	Hypocaloric LFD + PA	Histological NASH	52 weeks	↓	Associations between weight loss and histological improvement
Misciagna et al. (2017) [68]	RCT	44 vs. 46	Med. vs. CD	Moderate-severe (US)	6 months	Improvement	Significant improvement of NAFLD score
Abenavoli et al. (2017) [69]	RCT	20 vs. 20 vs. 10	Med. ± Antioxidant supplementation (1400–1600 kcal/d) vs. CD	Overweight	6 months	↓ of insulin resistance and fasting glucose	↓ of FLI and LSM in both diets
Markova et al. (2017) [70]	RCT	18 vs. 19	Isocaloric animal-protein vs. plant-protein diet	T2DM	6 weeks	↓ of insulin resistance and fasting glucose	48% vs. 35.7% ↓ in IHLC
Katsagoni et al. (2018) [71]	RCT	21 vs. 21 vs. 21	Hypocaloric Med. vs. Med. + lifestyle intervention vs. CD	Overweight/obese	6 months	Not evaluated	↓ of LSM in both diets, improvement in ALT only in Med. + lifestyle intervention-group
Marin-Alejandre et al. (2019) [72]	RCT	37 vs. 39	Hypocaloric diet vs. CD	Overweight/obese	6 months	Significant reduction in glucose and insulin	↓ in IHLC + FLI following both diets
Gepner et al. (2019) [73]	RCT	76 vs. 63 vs. 73 vs. 66	LFD vs. LFD with PA vs. Med./LCD vs. Med./LCD with PA	Abdominal obesity	18 months	Significant reduction in glucose and insulin	7.3% (Med./LCD) vs. 5.8 (LFD) ↓ in IHLC after 6 months, 4.2% vs. 3.8% after 18 months
Yaskolka Meir et al. (2020) [74]	RCT	89 vs. 84 vs. 91	Hypocaloric Med. (1500–1800 kcal/d ♂, 1200–1400 kcal/d ♀ vs. healthy diet	Abdominal obesity	18 months	Not evaluated	↓ IHLC following all diets
Xu et al. (2020) [75]	RCT	10 vs. 9 vs. 10	Hypocaloric LPD vs. HPD vs. reference-protein diet	Obese	3 weeks	Not evaluated	36.7% vs. 42.6% ↓ in IHLC vs. no changes in IHLC
Goss et al. (2020) [76]	RCT	14 vs. 11	LCD vs. LFD	Obese	8 weeks	↓ of insulin resistance	No significant difference
Holmer et al. (2021) [52]	RCT	20 vs. 24 vs. 20	LCD vs. 5:2 diet vs. CD	NAFLD	12 weeks	↓ of insulin resistance and HbA1c	53.1% vs. 50.9% vs. 16.8% ↓ in IHLC, 61.9% vs. 63.8% vs. 20.2% ↓ in CAP, change in IHLC 3.9% greater in LCD compared to CD and 2.6% in 5:2 diet compared to CD, ↓ in LSM in 5:2 diet and CD compared to LCD

CAP: controlled attenuation parameter, CD: control diet, NAFLD: non-alcoholic fatty liver disease, NASH: non-alcoholic steatohepatitis, LCD: low-carbohydrate diet, LFD: low-fat diet, PA: physical activity, HPD: high-protein diet, LPD: low-protein diet, LSM: liver stiffness measurement, IHLC: intrahepatic lipid content, LFD: low-fat diet, Med.: Mediterranean diet, FLI: fatty liver index, ↓: decrease, ♂: male, ♀: female

### 2.2. Cutting-Edge Solutions: Exploring Surgical Therapies for NAFLD

The potential effects of bariatric surgery on liver fat disease may extend beyond weight loss. In fact, serum concentrations of glucagon-like peptide-1 (GLP-1) increase after metabolic surgery, leading to decreased appetite, slower gastric emptying, and improved insulin sensitivity [77]. Furthermore, the main role of GLP-1 is to modulate bile acid signaling through the farnesoid X receptor (FXR), which can modify the gut microbiome and promote NAFLD [78]. Therefore, current guidelines recommend that metabolic surgery can be a potential approach in patients with T2DM or overweight/obese individuals (i.e., BMI > 35 kg/m^2^) [48,79]. Although bariatric surgery has a beneficial effect, various limitations such as patient acceptability of complications, availability of services, and high cost make its use difficult and highlight the need to carefully select eligible candidates [80].

The most common bariatric surgery procedures include adjustable gastric band (AGB), biliopancreatic diversion (BPD), vertical sleeve gastrectomy (SG) and Roux-en-Y gastric bypass (RYGB). Consequently, different methods might induce variable biological effects depending on the surgical procedure. The most common metabolic operations are SG and RYGB. In the first, about 80% of the stomach portion is removed along the gastric greater curvature and the small dimensions of the stomach, along with the changes in the hormonal environment, reduce hunger and delay gastric emptiness. In the second procedure, the stomach is separated into a smaller pouch in the smaller curvature (through stapling) and anastomosed with the jejunum [81,82]. In fact, by restricting food intake and by promoting malabsorption of nutrients, these techniques can cause weight loss. Both the reduction in body weight and decrease in waist circumstance through bariatric surgery led to improvement in insulin resistance, T2DM, obesity, fatty liver disease, and dyslipidemia [80]. Interestingly, one of the most important outcomes of bariatric surgery is that it can markedly improve all histological characteristics of NAFLD, including fibrosis. According to Lee et al., a resolution of steatosis was observed in 66% of patients, a resolution of inflammation in 50% of patients, and a resolution of fibrosis in 40% of patients after bariatric surgery [83]. Another recent study published in 2018, showed a resolution of NAFLD of 78% after RYGB [84]. Furthermore, Weiner reported that patients after AGB, RYGB and BPD achieved complete regression of NAFLD in up to 82.8% of the cases [85].

Table 3 presents studies that have investigated the effectiveness of bariatric surgery management in alleviating NAFLD. 

### 2.3. Exploring the Innovative World of Pharmaceutical Solutions

#### 2.3.1. Effects of Anti-Diabetic Agents on NAFLD

It is well established that fatty liver and T2DM are the two sides of the same coin, sharing common pathogenic pathways and factors such as insulin resistance. Although the coexistence of fatty liver and T2DM is increasing, the treatment is not adequate. Due to this close link between T2DM and NAFLD, various glucose-lowering drugs have been used as NAFLD therapeutics. Numerous clinical trials have shown the beneficial effects of GLP-1 receptor agonists, insulin-sensitizing thiazolidinediones, and sodium-glucose cotransporter 2 (SGLT2) inhibitors on liver fat content. GLP-1 binds to a specific GLP-1 receptor, whose activation can promote the reduction of liver steatosis by improving insulin signaling pathways, hepatocyte lipotoxicity, and mitochondrial function [95,96]. On the contrary, SGLT2 inhibition promotes negative energy balance through increased glycosuria and a change of the substrate to lipids as an energy source, which inhibit liver steatosis, inflammation, and fibrosis [97]. In 2021, a meta-analysis by Mantovani et al. showed that treatment with GLP-1 receptor agonists or SGLT2 inhibitors compared to placebo decreased the absolute percentage of liver fat content and serum levels of ALT [98]. These findings have been replicated by several other studies [99].

It is significant to mention that a multitude of studies have recently elucidated the efficacy of SGLT2 inhibitors in alleviating liver steatosis. Of particular interest, in 2018, Shibuya et al. revealed, for the first time, the statistically significant and advantageous impact of luseogliflozin, compared to metformin, on NAFLD and weight loss [100]. The results manifested a superior effect of the former, particularly on body mass index (BMI), after a six-month period. Luseogliflozin facilitates the mitigation of visceral adiposity by excreting energy through urine glucose excretion [100]. Additionally, in the same year, a double-blind randomized controlled study revealed that dapagliflozin monotherapy can reduce the levels of hepatocyte injury biomarkers, such as ALT, AST, γ-glutamyl transferase (γ-GT), cytokeratin 18-M30, cytokeratin 18-M65, and plasma fibroblast growth factor 21 (FGF21). With the combination of omega-3 carboxylic acids, not only can glucose control be improved, but body weight and abdominal fat volumes can also be reduced [101]. Analogous results were observed in a study by Sattar et al. using empagliflozin, in which the decline in ALT was consistently more notable than that of AST. The reductions were most prominent in participants with the highest baseline ALT levels and were primarily unaffected by changes in HbA1c and body weight [102]. However, the exact mechanisms through which empagliflozin mitigates aminotransferases or liver fat remain ambiguous; therefore, more research is needed.

Thiazolidinediones increase peripheral insulin sensitivity by stimulating adipokines, promoting triglyceride storage in adipose tissue, and improving the suppressive action of insulin on lipolysis [103]. In this way, thiazolidinediones lead to lower serum levels of free fatty acids and reduced hepatic lipid accretion [103]. The most common and approved are pioglitazone and rosiglitazone, which are powerful activators of the nuclear receptor PPARγ and are expressed mainly in adipose tissue [104]. Numerous studies have shown that pioglitazone has a beneficial effect on insulin sensitivity, inflammation, and hepatocyte degeneration, but there was no difference in the resolution of fibrosis [105]. A recent meta-analysis of randomized controlled trials using pioglitazone or rosiglitazone revealed that both drugs led to enhanced liver histology, including decreased steatosis, inflammation, and hepatocyte degeneration [106].

Recently, semaglutide and tirzepatide have been added to the therapeutic quiver against T2DM and obesity. Semaglutide is a GLP-1 analogue, and tirzepatide is a dual analogue of GLP-1 and GIP (glucose-dependent insulinotropic polypeptide) [107]. Both delivered impressive results in the phase 3 trials and can be considered future game changers in the realm of T2DM remission. Due to their importance in NASH therapy, these agents will be discussed in detail at a later part of the review. Table 4 summarizes the clinical trials that have examined the effects of antidiabetic drugs on NAFLD.

#### 2.3.2. Effects of Statins and Other Lipid-Lowering Drugs

Realizing that NALFD is strongly related to metabolic syndrome, there is a need for an integrated approach for individuals with a high liver fat content. The definition of metabolic syndrome includes the following criteria: abdominal obesity, with waist circumference of ≥90 cm in men or ≥85 cm in women; low high-density lipoprotein (HDL) cholesterol with HDL-cholesterol of <40 mg/dL in men and <50 mg/dL in women; hypertriglyceridemia with triglyceride (TG) of ≥150 mg/dL; high systolic blood pressure (BP), with systolic BP of ≥130 mmHg and/or diastolic BP of ≥85 mmHg; or hyperglycemia, with fasting plasma glucose (FPG) of >100 mg/dL [125,126]. Consequently, abnormal blood cholesterol levels play a key role in the progression of NAFLD and can be controlled with statins [127]. Except for this action, statins exhibit pleiotropic properties, such as antioxidant and anti-inflammatory effects, neoangiogenesis, and improvement of endothelial functions [127]. Interestingly, a large amount of data recommended that a statin remedy is correlated with a significant improvement in liver steatosis, inflammation, and even fibrosis [128,129]. For example, an observational study by Lee et al. found a lower risk of NAFLD in patients who received statin therapy [129]. According to a meta-analysis of six studies, ezetimibe, which is a lipid lowering agent acting by reducing cholesterol absorption in the intestines, significantly reduced plasma liver enzyme levels, as well as improved liver steatosis [130]. On the contrary, fenofibrate, a PPAR-a agonist, does not have a significant effect on liver fat content [131].

Furthermore, the category of omega-3 polyunsaturated fatty acids (n-3 PUFAs), such as a-linolenic acid (a-ALA), stearidonic acid (SDA), eicosapentaenoic acid (EPA), docosapentaenoic acid (DPA) and docosahexaenoic acid (DHA), has a beneficial effect on peripheral insulin sensitivity and on triglycerides levels, leading to a lower deposition of liver fat [132]. A randomized control trial published in 2020 reported lower liver fat in patients who received n-3 PUFA supplementation compared to those who received the placebo [132]. Similar outcomes were observed in another meta-analysis [133]. However, more well-designed randomized clinical trials are necessary to suggest omega-3 PUFA supplementation for the treatment of NAFLD in patients with and without T2DM.

Although statins have long been used as a widely accepted method for reducing cholesterol and minimizing the risk and mortality associated with cardiovascular disease, recent years have seen increasing scrutiny of their potential side effects. It should be noted, in particular, the increasing evidence linking prolonged statin use with diabetes progression and ectopic fat deposition, particularly in the kidneys of patients with diabetic nephropathy [134]. Recently, a study by Huang et al. revealed that statin administration for a period of more than 10 years was found to increase insulin resistance, alter lipid metabolism, provoke inflammation and fibrosis, and ultimately exacerbate the progression of diabetic nephropathy in diabetic mice. It is worth noting that the duration of the study spanned 50 weeks, which is equivalent to at least 35 years in the human life cycle [134]. Similar results were reported in the retrospective cohort study by Mansi et al., which followed patients with diabetes for a 12-year period and highlighted the need to carefully consider the metabolic effects of statin use when evaluating its risk–benefit ratio for diabetic individuals [135]. However, these findings are in stark contrast to the beneficial effects of statins in the treatment of NAFLD. Future research efforts are expected to provide more information on whether prolonged statin use may yield more detrimental than advantageous outcomes.

#### 2.3.3. From Lab to Liver: The Promising Future of Developing New Drugs for NAFLD

More recently, new potential drugs for NAFLD have been studied. The most common treatment targets are the farnesoid X receptor (FXR), a nuclear receptor, and obeticholic acid (OCA), which is a synthetically modified analogue of chenodeoxycholic acid [136]. These agents improve insulin resistance, regulate glucose and lipid metabolism, and have anti-inflammatory and anti-fibrotic effects in NAFLD [137]. Yonoussi et al. observed that 308 patients who received OCA 25 mg daily had an improvement in fibrosis, compared to the control group [138]. Interestingly, the multicenter, randomized, placebo-controlled FLINT trial documented that those who responded to OCA, defined as patients with a ≥30% reduction in liver lipid content, had an improvement in the histological features of NASH, including fibrosis and steatosis as well as reduced cell death [139]. However, one of the limitations of OCA use is that it can increase serum LDL levels. Therefore, the safety of the combination of OCA with statin in patients with NASH is being tested in ongoing studies. Non bile acid farnesoid X-activated receptor (FXR) agonists, including tropifexor, cilofexor, EDP-305, and nidufexo, have also been tested for NAFLD [140]. The main difference between OCA and cilofexor is that the latter caused a reduction in liver lipid content in patients with NASH without altering blood levels of lipids or indicators of insulin resistance [141]. An anti-inflammatory drug, the stearoyl-CoA desaturase (SCD) 1 inhibitor, plays a key role in liver lipogenesis. The main action is to catalyze the conversion of saturated fatty acids to MUFA, protecting against hepatic steatosis [142,143]. Two animal studies showed that SCD1 activity was elevated in proportion to liver lipid content in models of NAFLD and genetic knockout of hepatic SCD1 expression effectively reduced fatty liver and insulin resistance in animals fed a high-fat diet [144,145]. Recently, a clinical trial revealed a reduction in liver fat content, resolution of NASH, and improvement of liver fibrosis in individuals with NAFLD and prediabetes or T2DM after the use of aramchol [146]. Given that to date there is no FDA-approved therapy for the treatment of NAFLD, evaluating the efficacy and safety profile of new agents is of paramount importance. However, the fact that liver biochemistry does not always reflect hepatic histology makes the conduction of biopsy studies important, which on the other hand, present significant technical challenges for obvious reasons.

Table 5 presents ongoing trials investigating the safety and efficacy of new medications for the treatment of NAFLD.

#### 2.3.4. Effects of Anti-Obesity Drugs

During recent years, weight loss pharmacotherapy has been an attractive option for patients with NAFLD, with or without T2DM and with a BMI > 30 kg/m^2^ or >27 kg/m^2^ in the presence of at least one metabolic comorbidity, because it can lead to the remission of diabetes and improves the progression of fatty liver disease [143,146]. Τhe US Food and Drug Administration (FDA) has approved six medications for chronic weight management: orlistat, lorcaserin, phentermine/topiramate, bupropion/noltrexone, liraglutide and semaglutide that are associated with a decrease in body weight of at least 5% in one year [146]. Interestingly, a closer look at the literature reveals that, except for the long-term effects on glycemic control, orlistat selectively reduces visceral fat and prevents the digestion of free fatty acids that are responsible for the increase in liver and peripheral insulin resistance [147]. In 2005, a meta-analysis of seven randomized control trials showed that patients who received 120 mg orlistat three times received an average weight loss of 3.8 kg compared to 1.4 kg in the placebo group after 12 weeks [148]. In contrast, no data are available on the effects of topiramate, naltrexone, bupropion, and phentermine on liver outcomes in patients with NAFLD [149].

## 3. Tirzepatide and Semaglutide: New Weapons against NAFLD

As previously stated, two recently approved drugs, tirzepatide, a dual analogue of GIP/GLP-1, and semaglutide, a GLP-1 analogue, were added to the therapeutic arsenal of NAFLD [150]. A phase 3 trial reported that the liver fat content was significantly reduced after tirzepatide therapy compared to baseline measures [151]. On the other hand, a phase 2 trial involving patients with NASH revealed that semaglutide treatment resulted in a significantly higher percentage of patients with NASH resolution than the placebo [113].

Tirzepatide is associated with a significant reduction in liver fat content (LFC), subcutaneous abdominal adipose tissue (ASAT), and volume of visceral adipose tissue (VAT) [152,153]. A multicenter phase 3 clinical trial reported a significant reduction in LFC, ASAT, and VAT volumes compared to insulin degludec in individuals with T2DM [151]. The results were more prominent in groups receiving 10 mg and 15 mg of tirzepatide. An MRI scan was performed in participants before randomization and the primary efficacy endpoint was identified by MRI-proton density fat fraction (MRI-PDFF) at week 52 of drug administration [151]. According to a meta-analysis of 11 phase 2 randomized controlled trials, the treatment regimen with a GLP-1 receptor agonist (liraglutide or semaglutide) for a median of 26 weeks was strongly associated with significant reductions in LFC, as well as increased histological resolution of NASH without worsening of fibrosis, compared to the placebo or reference therapy group [154].

In 2020, Hartman et al. determined the result of tirzepatide on the biomarkers of NASH and fibrosis in people with diabetes. A reduction in ALT, AST, keratin-18 (K-18), and N-terminal pro-peptide of type III collagen (Pro-C3) levels was revealed, combined with a significant increase in adiponectin levels that was dose-dependent [155]. It is a well-established fact that K-18 and Pro-C3 serve as biomarkers of fibrogenesis in individuals afflicted with NAFLD. As mentioned above, to date, there has not been approved pharmacological therapy to resolve NAFLD. However, the remarkable effects of tirzepatide on LFC promise that it can be more effective than GLP-1 receptor agonists, but the preliminary promising findings should be confirmed by further studies. An ongoing trial, SYNERGY-NASH, will provide specific data on the beneficial effect of tirzepatide in participants with biopsy-proven NASH.

Semaglutide has also been effective in improving liver biochemistry. A prospective study by Volpe et al. showed a significant decrease in glucometabolic parameters, liver enzymes, and laboratory indicators of liver steatosis during therapy, as well as a reduction in fat mass and VAT. Interestingly, liver steatosis improved in 70% of participants after 52 weeks of treatment [156]. Furthermore, in 2021, a randomized control trial compared changes in liver stiffness and liver fat in individuals with NAFLD who received semaglutide and placebo therapy, respectively. The results revealed a significant reduction in liver steatosis with an improvement in liver enzymes and metabolic parameters in the semaglutide group versus the placebo group. However, no significant differences in liver stiffness were observed in both groups [157]. This year, Mantovani et al. published a systematic review providing data on the beneficial effect of GLP-1 receptor antagonists on the histological features of NAFLD, including steatosis, ballooning, and lobular inflammation, as well as on the resolution of NASH without worsening of fibrosis [158].

Recently, an open-label phase 2 trial by Alkhouri et al. reported that the combination of semaglutide with cilofexor and firsocostat achieved greater improvements in liver steatosis and biochemistry profile than monotherapy [159]. Based on the evidence mentioned above, semaglutide in combination with weight loss and a specific diet should be recommended as an optimal approach for the treatment of individuals with T2DM and fatty liver disease.

Future studies are expected to add additional evidence on the effects of new antidiabetic agents on NAFLD progression and clarify whether the benefits are driven primarily by weight loss or by the pleiotropic actions of the drugs.

## 4. Conclusions

The review of available evidence supports the notion that NAFLD is an important risk factor for the development of T2DM, but also underlines the bidirectional relationship between these conditions. Despite many advances in our knowledge on the epidemiology and pathogenesis of NAFLD, the most established treatment so far is intensive weight loss. The importance of weight reduction is highlighted in people with NASH, where weight loss greater than 7% is associated with a clinically significant regression of disease status [45]. Exercise compared to weight loss produces significant but modest changes in liver fat. Lifestyle modification, including specific diets and physical activity, is and should be the first line of treatment in NAFLD and NASH because it has been shown to ameliorate the risk of extrahepatic comorbidities and complications. On this basis, the current European and American clinical guidelines for the treatment of NAFLD have strongly suggested the importance of routine screening for T2DM in all patients with NAFLD [160].

Today, the question is no longer whether lifestyle modification is an effective clinical therapy, but how we implement lifestyle as a therapy in daily clinical care. However, most studies indicate several barriers to maintaining a healthy lifestyle in the long term and that weight regain is very common among people with diabetes and/or obesity, making the use of pharmacotherapy a necessity in most cases. In this context, a better understanding of the mechanisms linking NAFLD and T2DM will help design targeted therapies that can stop or reverse disease progression.

## Figures and Tables

**Table 1 metabolites-13-00517-t001:** Effects of exercise in patients with NAFLD.

Study	Number of Patients	Type of Exercise	Duration of Intervention	Outcomes
Goodpaster et al. (2010) [37]	*n* = 130	Aerobic Exercise with diet	6 months vs. 12 months	↓ hepatic lipid content in 12 months, equal ↓ of insulin resistant
Finucane et al. (2010) [38]	*n* = 100	Aerobic Exercise vs. control	12 weeks	↓ hepatic lipid content
Slentz et al. (2011) [39]	*n* = 196	Aerobic Exercise vs. Resistance Exercise vs. Aerobic + Resistance Exercise	8 months	Groups including aerobic exercise: higher ↓ hepatic fat content, ↓ ALT, and ↓ insulin resistance
Jakovljevic et al. (2013) [33]	*n* = 17	Resistance training vs. Control	8 weeks	↓ hepatic fat contents in resistance training group
Wong et al. (2013) [40]	*n* = 154	Aerobic Exercise vs. Control	12 months	↓ hepatic fat content
Zelber-Sagi et al. (2014) [41]	*n* = 82	Resistance Exercise vs. Control	12 weeks	Improving of steatosis and inflammation
Balducci et al. (2015) [42]	*n* = 606	Aerobic Exercise + Resistance Exercise vs. Control	12 months	↓ fatty liver index
Cuthbertson et al. (2016) [43]	*n* = 69	Aerobic Exercise vs. Control	16 weeks	↓ hepatic lipid content, ↑peripheral insulin sensitivity
Zhang et al. (2016) [32]	*n* = 220	Aerobic Exercise vs. control	12 months	↓ hepatic fat content
Skrypnik et al. (2016) [44]	*n* = 44	Aerobic Exercise vs. Aerobic + Resistance Exercise	3 months	Greater reduction in ALT and AST
Oh et al. (2017) [31]	*n* = 61	Aerobic + Resistance training	12 weeks	↓ ALT, ↓ AST and ↓ TG
Farzanegi et al. (2018) [45]	*n* = 49 (rats)	Aerobic training	4 weeks	↓ hepatic cell apoptosis, ↓ ALT, ↓ AST and ↓ ALP

AST: aspartate aminotransferase, ALT: alanine aminotransferase, TG: triglycerides, ALP: alkaline phosphatase, ↓: decrease.

**Table 3 metabolites-13-00517-t003:** The effectiveness of bariatric surgery management in alleviating NAFLD.

Study	Number of Patients	Type of Surgery	Outcomes
Weiner et al. (2010) [85]	*n* = 116	RYGB, AGB, BPD	Complete regression of NAFLD in 82.8%
Moretto et al. (2012) [86]	*n* = 78	RYGB	Improvement of fibrosis from 44.8% to 30.8%
Tai et al. (2012) [87]	*n* = 21	RYGB	↓ of steatosis, NASH, and fibrosis
Vargas et al. (2012) [88]	*n* = 26	RYGB	↓ of steatosis, NASH, and fibrosis
Taitano et al. (2014) [89]	*n* = 160	RYGB, AGB	75% resolution of steatosis, 90% resolution of NASH, 50% resolution of fibrosis
Lassailly et al. (2015) [90]	*n* = 109	RYGB, AGB, BIB	Resolution of NASH in 85% of patients Reduction of fibrosis in 34% of patients
Aldoheyan et al. (2017) [91]	*n* = 27	RYGB, BPD	Improvement of steatosis and fibrosis
Parker et al. (2017) [92]	*n* = 37	RYGB	RYGB reverses NASH and fibrosis
Esquivel et al. (2018) [93]	*n* = 43	SG	100% improvement of NAFLD
Schwenger et al. (2018) [84]	*n* = 42	RYGB	78% resolution of NAFLD 9.5% worsening of fibrosis
Pooler et al. (2019) [94]	*n* = 50	RYGB	Improvement of steatosis

NAFLD: non-alcoholic fatty liver disease, NASH: non-alcoholic steatohepatitis, RYGB: Roux-en-Y gastric bypass, SG: sleeve gastrectomy, AGB: adjustable gastric banding, BIB: biliointestinal bypass, BPD: biliary-pancreatic diversion, ↓: decrease.

**Table 4 metabolites-13-00517-t004:** Beneficial effects of anti-diabetic drugs on NAFLD.

Study	Number of Patients	Duration	Study Population	Drug Name	Dose	Liver Outcomes	Diabetes Outcomes
Dutour A et al. (2016) [108]	*n* = 44	26 weeks	Obesity, T2DM, NAFLD	Exenatide	5 μg twice daily 4 weeks and after 10 μg/day	↓ Hepatic triglyceride	↓ Weight
Armstrong MJ et al. (2013) [109]	*n* = 4442	Meta-analysis	Obesity, T2DM	Liraglutide	1.8 mg/day	↓ ALT	Not mentioned
Armstrong MJ et al. (2016) [110]	*n* = 7	12 weeks	Obesity, NASH	Liraglutide	1.8 mg/day	↓ ALT, AST, DNL	↓Weight and adipose mass, ↓ HbA1c and serum levels of glucose, ↓ LDL cholesterol, ↑ insulin sensitivity
Armstrong MJ et al. (2016) [111]	*n* = 26	48 weeks	Obesity, NASH	Liraglutide	1.8 mg/day	↑ NASH resolution	↓ Weight, ↓ HbA1c and serum levels of glucose, ↑ HDL cholesterol
Newsome P et al. (2019) [112]	*n* = 957	Meta-analysis	Obesity, with or without T2DM	Semaglutide	Escalation from 0.05 to 0.4 mg/day	↓ ALT	Not mentioned
Newsome P et al. (2021) [113]	*n* = 320	72 weeks	Obesity, NASH (F1–F3 fibrosis)	Semaglutide	0.1, 0.2, or 0.4 mg/day	↑ NASH resolution, ↓ ALT, AST	↓ Weight, ↓ HbA1c
Cui J et al. (2016) [114]	*n* = 25	24 weeks	Obesity, pre-diabetes, early T2DM	Sitagliptin	100 mg/day	No changes	No changes
Joy TR et al. (2017) [115]	*n* = 12	24 weeks	Obesity, T2DM, NASH	Sitagliptin	100 mg/day	No changes	No changes
Neuschwander-Tetri BA et al. (2003) [116]	*n* = 30	48 weeks	Obesity, NASH	Rosiglitazone	4 mg twice daily	↓ ALT and AST, ↓ steatosis and inflammatory	↓ HbA1c and fasting insulin, ↑ glucose tolerance, ↑ weight
Ratziu V et al. (2008) [117]	*n* = 63	52 weeks	Obesity, NASH	Rosiglitazone	4 mg/day 1 month and after 8 mg/day	↓ ALT, ↓ steatosis	↓ Fasting glucose and insulin, ↑ weight
Belfort R et al. (2006) [104]	*n* = 55	24 weeks	Obesity, T2DM, NASH	Pioglitazone	45 mg daily	↓ ALT and AST, ↓ steatosis and inflammatory	↓ Fasting glucose and insulin, ↑ insulin sensitivity, ↑ HDL cholesterol, ↑ weight
Cusi K et al. and Sanyal AJ et al. (2016 + 2010) [118,119]	*n* = 101, *n* = 247	96 weeks	Obesity, NASH	Pioglitazone	45 mg/day, 30 mg/day	↓ ALT and AST, ↓ NAS, ↑ NASH resolution	↓ Fasting glucose, insulin and triglyceride, ↑ insulin sensitivity, ↑ HDL cholesterol, ↑ weight
Cusi K et al. (2019) [120]	*n* = 56	24 weeks	Obesity, T2DM	Canagliflozin	300 mg/day	No changes	↓ Weight, ↓ HbA1c, fasting glucose and insulin, ↑ hepatic insulin sensitivity
Latva-Rasku A et al. (2019) [121]	*n* = 32	8 weeks	Obesity, T2DM	Dapagliflozin	10 mg/day	↓ Liver lipids, ↓ liver stiffness	↓ Weight, ↓ HbA1c, fasting glucose
Shimizu M et al. (2019) [122]	*n* = 57	24 weeks	T2DM, NAFLD	Dapagliflozin	5 mg/day	↓ ALT	↓ Weight
Kahl S et al. (2020) [123]	*n* = 42	24 weeks	Obesity, T2DM	Empagliflozin	25 mg/day	↓ Liver lipids	↓ Weight, ↓ glucose
Kuchay MS et al. (2018) [124]	*n* = 25	20 weeks	T2DM, NAFLD	Empagliflozin	10 mg/day	↓ ALT, ↓ liver lipids	No changes

T2DM: type 2 diabetes mellitus, HbA1c: glycated hemoglobin, NAFLD: non-alcoholic fatty liver disease, NASH: non-alcoholic steatohepatitis, NAS: NAFLD Activity Score, AST: aspartate aminotransferase, ALT: alanine aminotransferase, HDL: high-density lipoproteins, LDL: low-density lipoproteins, ↓: decrease, ↑: increase.

**Table 5 metabolites-13-00517-t005:** Ongoing trials of new medications for the treatment of NAFLD.

Drug	Trial Identifier	Number of Patients	Mechanism	T2DM Inclusion Criteria	Primary Endpoint
Elafibranor	NCT02704403	2000	PPARα/δ dual agonist	T2DM only with HbA1c ≤ 9%	% of patients with NASH resolution without fibrosis worsening at week 72 from BL, long term liver-related outcomes
Saroglitazar	NCT04193982	250	PPAR-α/γ agonist	NA	Change in NFS at week 8, 16, and 24
Obeticholic Acid	NCT03439254	919	FXR agonist	T2DM only with HbA1c ≤ 9.5%	% of patients with improvement of liver fibrosis by ≥1 stage with no worsening of NASH after 18 months
Obeticholic Acid	NCT02548351	2480	FXR agonist	T2DM only with HbA1c ≤ 9.5%	Improvement of liver fibrosis by ≥1stage with no worsening of NASH OR achieving NASH resolution without worsening of liver fibrosis at month 18 from BL, long term liver-related outcomes
Cenicriviroc	NCT03028740	2000	CCR2/5 dual antagonist	T2DM with HbA1c ≤ 10%	Improvement of liver fibrosis by ≥1 stage with no worsening of NASH after 12 months, long term liver-related outcomes
Aramchol	NCT04104321	2000	SCD1 inhibitor	T2DM with controlled glycemia or prediabetes	NASH resolution with no worsening of fibrosis OR fibrosis improvement by ≥1 stage with no worsening of NASH at week 52 from BL, Long term liver-related outcomes
Resmetirom	NCT03900429	2000	THR-β agonist	T2DM with HbA1c < 9%	NASH resolution in patients with F2-F3 fibrosis after 52 weeks, long term liver-related outcomes

BL: baseline, CCR2/5: C-C chemokine receptors type 2 and type 5, FXR: farnesoid X receptor, HbA1c: glycated hemoglobin, LXR: liver X receptor, NA: data not available, NAFLD: non-alcoholic fatty liver disease, NFS: NAFLD fibrosis score, PPAR: peroxisome proliferator-activated receptor, NASH: non-alcoholic steatohepatitis, SCD: stearoyl-CoA desaturase, SGLT: sodium-glucose cotransporter, THR: thyroid hormone receptor, T2DM: type 2 diabetes.

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
