# Peer review of "Therapeutic Potentials of Reducing Liver Fat in Non-Alcoholic Fatty Liver Disease: Close Association with Type 2 Diabetes"

_metabolites, 2023, doi:10.3390/metabo13040517_

Round 1
Reviewer 1 Report
The authors covered most of the perspectives that cover the role of reducing excessive liver fat in prevention or management of T2D. However, for more comprehensive review, The following points are highly recommended to be incorporated:
11. In page 13 (2.3.2. Effect of statins and other lipid lowering drug. It should be stated that : although statins are widely used for lowering cholesterol and reducing risk if cardiovascular disease, however, long term use of statins e.g. 10 years or longer by diabetic patients should be reassessed for the possibility of increased ectopic fat deposition in kidney and exacerbating diabetic nephropathy.
Ref.: Long-term statins administration exacerbates diabetic nephropathy via ectopic fat deposition in diabetic mice. Huang T. et al. 2023, Nature Communications | (2023) 14:390
This refences presented in introduction data from clinical trials in patients that this point is still controversial and supported this concern in mouse model of diabetes with long term use of statins (50 weeks equals at least 35 years in human life cycle).
2. Although manuscript covered dietary interventions, yet the nutritional intervention was missed.
Dietary relates mainly to caloric restrictions and various levels of fats vs. carbohydrates in diet composition.
But there are several studies recommended certain nutrients for intervention to control level of fatty liver and improve T2D and insulin response such as apple vinger, green tea, olive oil, and lemon Therefore, I highly suggest adding a separate sub-title for: reducing fatty liver and improving T2D by nutritional interventions, where these interventions can be summarized.
Down are some main example references to be considered.
Beneficial Effects of Apple Vinegar on Hyperglycemia and Hyperlipidemia in Hypercaloric-Fed Rats. Driss Ousaaid, Hassan Laaroussi, [...], and Ilham ElArabi. J Diabetes Res. 2020:2020:9284987
Antihyperglycemic, Antihyperlipidemic and Modulatory Effects of Apple Cider Vinegar on Digestive Enzymes in Experimental Diabetic Rats.
Ben Hmad Halima, Khlifi Sarra, Ben Jemaa Houda, Gara Sonia and Aouidet Abdallah. International Journal of Pharmacology, 2016, Volume:12, Issue :5, page:505-513.
Oily fish, coffee, and walnuts: Dietary treatment for nonalcoholic fatty liver disease. Vikas Gupta, Xian-Jun Mah, Maria Carmela Garcia, Christina Antonypillai, David van der Poorten.
World Journal of Gastroenterology 2015 October 7; 21(37): 10621-10635
Reviewer 2 Report
The authors present a comprehensive review of the relationship between NAFLD and T2DM and the pharmacological and non-pharmacological interventions that can be used in these metabolic disorders. The manuscript is written clearly and I have no major comments. In line 337, please provide a description for K-18 and Pro-C3.
Reviewer 3 Report
In this review, Tsamos et al. summarize the association of NAFLD with T2D and potential treatments. This review is well written and organized with full of useful information for readers in the field. I do not have comments for this manuscript except for the title. The main text mentions predominantly about NAFLD, including potential treatments and trials. The association between NAFLD and T2D is minimal in this review, so title should highlight NAFLD, not T2B, such as: Therapeutic potentials of reducing fat in non-alcoholic fatty liver disease: Close association with type 2 diabetes.
